# Acute Abdominal Pain with Obstructing Trichobezoar: A Pediatric Case of Rapunzel Syndrome Diagnosed in a Pediatric Emergency Department

**DOI:** 10.3390/pediatric17030053

**Published:** 2025-04-30

**Authors:** Sung-Ha Kim, Jong-In Lee, Soohyun Park, So-Hyun Paek

**Affiliations:** 1Department of Emergency Medicine, CHA Bundang Medical Center, CHA University School of Medicine, Seongnam 13497, Republic of Korea; 2Department of Pediatric Surgery, CHA Bundang Medical Center, CHA University School of Medicine, Seongnam 13497, Republic of Korea

**Keywords:** abdominal pain, Rapunzel syndrome, small bowel obstruction, trichophagia, trichobezoar

## Abstract

Introduction: Acute abdominal pain is a prevalent complaint in pediatric emergency departments. Primary care physicians can find it challenging to treat such pain and, in particular, to determine whether it requires emergent surgical intervention. Acute appendicitis is the most common surgical cause of abdominal pain, but it is important to understand that other rare conditions can also be life-threatening. Case presentation: We report the case of a 6-year-old girl who presented to our pediatric emergency center with complaints of acute abdominal pain, vomiting, and diarrhea. She had no notable medical history, including perinatal, surgical, or psychiatric disorders. After finding a bezoar-like structure through a combined enteritis CT scan, reassessing the child’s dietary concerns revealed that the child had experienced symptoms of trichophagia for approximately 3 to 4 years. Enterotomy and the removal of the bezoar were successfully performed. A pediatric psychiatric consultation was carried out to prevent further trichophagia-induced complications. Conclusions: Despite our patient’s relatively young age and the failure to obtain a history of trichophagia at the onset, we successfully diagnosed a rare condition called Rapunzel syndrome. Although several cases of this condition have been reported by pediatric surgeons, we emphasize the role of physicians in pediatric emergency departments when examining children with this rare syndrome.

## 1. Introduction

Acute abdominal pain is a prevalent complaint in pediatric emergency departments (EDs). Primary care physicians can find it challenging to treat such pain and, in particular, to determine whether it requires emergent surgical intervention. The signs of abdominal conditions requiring surgical intervention include rebounding pain, involuntary guarding or rigidity, abdominal distention, and diffuse tenderness [1]. Despite evaluating all the features mentioned, the possibility of overlooking a serious etiology is a major concern for physicians in pediatric EDs. Acute appendicitis is the most common surgical cause of abdominal pain [2]; therefore, physicians tend to focus on certain physical indications, such as right-sided or rebounding tenderness. However, it is important to understand that other rare conditions can also be life-threatening.

Bezoars usually accumulate within the gastrointestinal tract, because they are made of foreign bodies that cannot be digested by gastric fluids [3]. A trichobezoar is a type of bezoar comprising hair, which is diagnosed almost exclusively in adolescent or young-adult women. The common clinical symptoms of a trichobezoar are a palpable tumor-like mass in the abdomen, abdominal pain, emesis or vomiting, weakness, weight loss, constipation or diarrhea, and hematemesis [4].

Herein, we report the case of a 6-year-old girl who presented to our pediatric emergency center with a chief complaint of acute abdominal pain and was diagnosed with Rapunzel syndrome. Although the condition is rare, several cases have been reported, particularly by pediatric surgeons [5]. We emphasize the role of physicians in pediatric EDs in examining children with this rare syndrome.

## 2. Case Presentation

A 6-year-old girl presented to our pediatric ED with a 2-day history of abdominal pain, vomiting, and diarrhea. She visited a local primary clinic and was referred to our center with suspected appendicitis or persistent gastroenteritis. Her caregiver stated that her oral intake had reduced, and she had been urinating less than usual. She was not taking any medication and had no notable medical history, including perinatal, surgical, and psychiatric disorders. The patient had no relevant family history of any other medical diseases. Her immunization status was up-to-date. No neurodevelopmental delay was observed. In addition, the caregiver mentioned that the child socially interacts and gets along with her peers fairly well during day care.

Upon physical examination, the patient’s vital signs were stable. She weighed 21 kg, which was approximately the 50th percentile of her age. She was 124 cm high, and her body mass index was 15, which was within the normal limits. Her general condition was slightly lethargic; however, the capillary refill time was within 2 s, and her tongue did not appear to be dehydrated. An abdominal examination revealed that her abdomen was soft and flat, and the bowel sounds seemed hypoactive. She experienced severe and unresolving right-sided periumbilical pain and tenderness on the right side of abdomen from the right upper quadrant to the right lower quadrant. We suspected appendicitis and gastroenteritis or colitis and immediately performed laboratory tests and computed tomography (CT) scanning.

Her complete blood count showed mild leukocytosis, other laboratory tests showed normal kidney and liver functions, and the urine analysis results were within the normal limits. Mild hypochloremia (96 mEq/L) and mildly elevated uric acid (6.6 mg/dL) levels were detected, which were considered the result of the child’s dehydrated state. The C-reactive protein level was elevated, at 1.05 mg/dL (the normal level is under 0.3 mg/dL).

The chest radiographic findings were within the normal limits. An abdominal radiography revealed prominent dilated bowel loops in the mid-abdomen with air-fluid levels, suggesting severe ileus or bowel obstruction (Figure 1). An abdominal CT scan was performed promptly in the ED, which showed diffuse small bowel dilatation with a small bowel feces-like structure at the distal ileum and mid-bowel wall enhancement, suggesting partial small bowel obstruction by a bezoar-like structure with combined enteritis. The appendix size was within the normal limits (Figure 2). After evaluating the CT scans, we carefully reassessed the child’s diet, habits, and previous psychiatric concerns. Initially, no history of trichophagia was mentioned when asked about any recent dietary concerns. However, when we tried to re-evaluate the medical history, the caregiver revealed that the child had experienced symptoms of trichophagia for approximately 3 to 4 years. This led us to suspect that the obstructive material was most likely a trichobezoar in the long segments of the ileal lumen.

First, intravenous hydration with isotonic fluid was performed immediately to resolve the child’s dehydration, and she fasted overnight for operative intervention. Exploratory laparotomy was performed the following day. A U-shaped trichobezoar was observed. It was approximately 20 cm in length and located in the distal ileum. Severe lymphadenitis was observed throughout the mesentery. Enterotomy and the removal of an identified bezoar were successfully performed (Figure 3). No other trichobezoars were identified. Further pathologic diagnostic examination was conducted with the single piece of irregularly shaped green bezoar, measuring 18.5 × 2.5 × 2.4 cm. The result was consistent with a trichobezoar.

A pediatric psychiatric consultation was conducted to prevent further trichophagia-induced complications. The patient was discharged on postoperative day 7 in good condition with outpatient clinic follow-up appointments for pediatric surgery and pediatric psychiatry. A week after discharge, she was examined in the pediatric surgeon’s outpatient clinic, and her recovery seemed uneventful. Cognitive behavior and developmental assessments were carried out on the same day, and regular follow-up checks were planned in the pediatric psychiatric department.

## 3. Discussion

The frequency of surgical intervention in children with acute abdominal pain is approximately 1% [6]. Acute appendicitis is the most common cause of emergency surgery [7]. Tseng et al. reported demographic and other related results in every pediatric patient who was admitted to the pediatric ED over 2 years. In their study, intussusception and incarcerated inguinal hernia were the major causes of abdomen surgery in children < 1 year of age, and acute appendicitis was the most common etiology of abdomen surgery in children > 1 year of age [8]. Intestinal obstruction accounts for approximately 15 percent of all emergency cases of acute abdominal pain [9]. Sudden obstruction is often associated with serious intra-abdominal conditions requiring immediate diagnosis and decision-making in the ED and may require prompt emergency surgical intervention. As we learned from previous studies and well-known medical facts, almost all etiologies of acute surgical abdomens in children can result in devastating complications if the diagnosis is misguided or treatment is delayed. In our case, we verified a very rare cause of pediatric abdomen surgery in the ED through history-taking and radiological evaluation. Considering previous studies, it is rare to pursue surgical intervention because of intestinal obstruction. However, the cause of the blockage was even rarer in our case. Our case also demonstrates the way in which pediatric emergency healthcare providers react to right lower quadrant pain. Because the symptoms of appendicitis in children can be atypical compared to those of adults, physicians in EDs sometimes fear failing to diagnose appendicitis. This may lead to more CT scans being unnecessarily performed in children with right lower quadrant pain [10]. We also reviewed our diagnostic flow and concluded that performing ultrasound before a CT scan would have been more helpful for detecting the bezoar.

Trichophagia is an eating disorder that is usually associated with trichotillomania. It is a chronic mental disease of impulse control, characterized by repetitive, compulsive, and self-induced hair pulling [11]. Ingested hair is slippery and, therefore, difficult to digest. Once the hair is entangled and accumulates between the mucosal folds of the stomach, it is called a trichobezoar. In patients with trichophagia, the incidence of gastric bezoars is approximately 0.5% [12]. Due to the demographic characteristics of its causal psychiatric entity, 90% of trichobezoar cases have been reported in young females aged 10 to 19 years old [13]. Our patient was younger than the age group previously reported to be the most affected.

The standard examination for trichobezoars is contrast-enhanced CT scanning. Ultrasonography may be helpful, but it is not fully pathognomonic, especially as air bubbles trapped between the hair limit its assessment capability [14]. A well-defined ovoid intraluminal heterogeneous mass with interspersed gas is the typical CT image of a gastric trichobezoar. If the bezoar passes the gastric outlet and causes an obstruction like in our case, there will be dilated intestinal loops in addition to the intraluminal mass with gas bubbles retained [15]. It would be much easier if the above radiologic characteristics were present, with a history of trichophagia or trichotillomania. In our case, the initial failure to note a history of trichophagia prevented a possibly quicker diagnosis. This demonstrates the importance of thorough history-taking by caregivers in the pediatric ED.

Bezoars often remain in the stomach; however, they can go through the gastric outlet and enter the small bowel, causing intestinal obstruction. This causes severe abdominal pain with obstructive symptoms such as malaise, vomiting, anorexia, and possibly weight loss and cachexia [16]. The so-called Rapunzel syndrome was first introduced and described by Vaughan et al. in 1968 [17]. Despite its rare presentation, several case reports and series have been published, particularly by pediatric surgeons. Mirza et al. shared their experience with 17 cases of gastrointestinal trichobezoars. Of the 17 patients, only 4 were found to have small bowel obstruction, as in our case, and 7 patients presented with a history of trichophagia [18]. Habib et al. reported six patients with trichobezoars in 2022. Among these cases, four presented to the ED, and three of them could not obtain any possible preoperative diagnosis of a trichobezoar. They stated that the variety of symptoms that the patients presented with was interesting. Some of them were completely unaware of their condition or any history of trichophagia and visited only because of abdominal pain or a mass [19]. Recently, Kaba et al. reported five cases, and two of them were diagnosed with Rapunzel syndrome; the youngest patient with gastric trichobezoar was a 5-year-old. In this study, all the patients were initially diagnosed with a history of trichophagia [20].

The primary goal of treating bezoars is their removal and preventing their recurrence. Several treatment options can be found in the literature, including removal through conventional laparotomy, laparoscopy, and endoscopy [21]. Mostly, surgical removal was performed via laparotomy [22]. Successful removals using the laparoscopic procedure have also been reported [23]. In our case, a laparotomy was performed. Our pediatric surgeon carefully reviewed the patient’s CT scan and then decided to perform a laparotomy. This decision was made because the trichobezoar was very long, which could have left undetected satellite trichobezoars or small tail-like parts of the trichobezoar in the small bowel.

In addition, most previous studies have emphasized that psychiatric evaluation and management should follow surgery. Cognitive behavioral therapy, along with other forms of psychotherapy, antidepressants, and antipsychotic medications, is known to enable successful prevention and recovery [24].

## 4. Conclusions

Despite our patient’s relatively young age and the failure to obtain a history of trichophagia at onset, we successfully diagnosed a rare condition called Rapunzel syndrome in the ED by carefully retaking the related medical history and conducting thorough radiological examinations. Our experience and other studies have shown that trichobezoar-related cases and Rapunzel syndrome require multidisciplinary diagnosis and treatment. A preoperative diagnosis by a pediatric emergency physician can be made by following a careful approach to determine the history of trichophagia and trichotillomania. Various radiologic methods, such as ultrasonography, radiography, and computed tomography (CT), are beneficial for preventing delayed diagnosis and complications. Medical and surgical interventions are needed if intestinal obstruction occurs, and psychiatric resolution must be achieved. Most previous studies have limitations due to the small number of patients assessed. Further studies with larger sample sizes and education are needed for healthcare providers in pediatric EDs.

## Figures and Tables

**Figure 1 pediatrrep-17-00053-f001:**
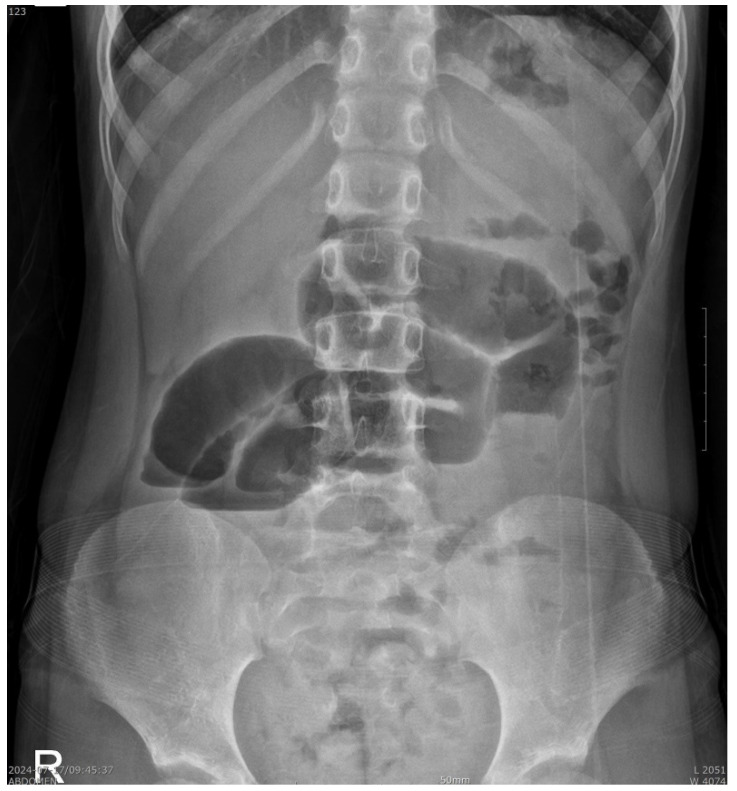
A plain radiograph showing prominent dilated bowel loops in the mid-abdomen with air-fluid levels, suggesting severe ileus or bowel obstruction.

**Figure 2 pediatrrep-17-00053-f002:**
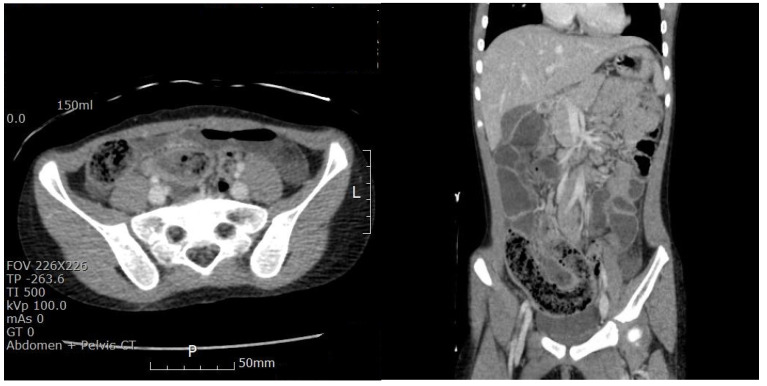
A computed tomography scan showing diffuse small bowel dilatation with a small bowel feces—like structure at the distal ileum and mid-bowel wall enhancement, suggesting small bowel obstruction by a bezoar-like structure with combined enteritis.

**Figure 3 pediatrrep-17-00053-f003:**
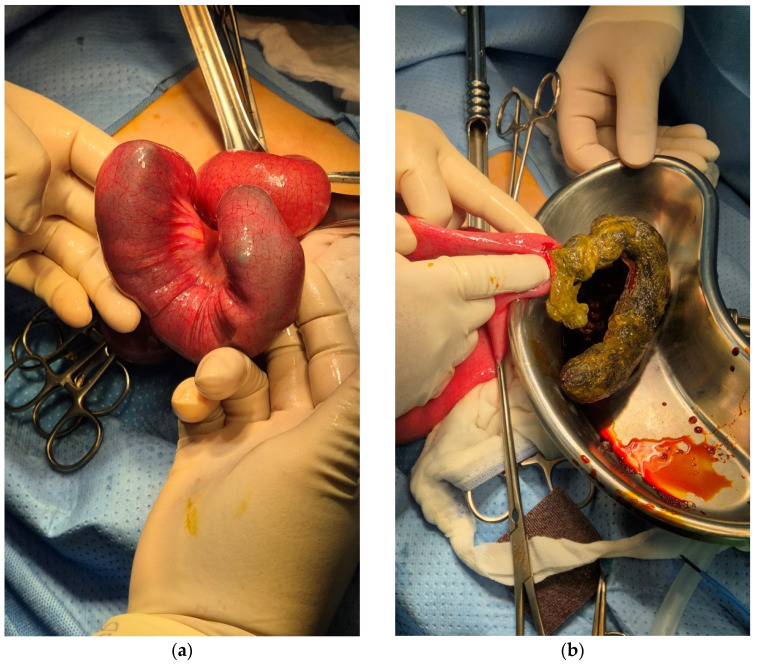
(**a**) A 20 cm U-shaped trichobezoar was found in the distal ileum; (**b**) the trichobezoar was removed after enterotomy.

## Data Availability

The data presented in this study are available on request from the corresponding author due to privacy restrictions.

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
