# Peer review of "Acute Abdominal Pain with Obstructing Trichobezoar: A Pediatric Case of Rapunzel Syndrome Diagnosed in a Pediatric Emergency Department"

_pediatrrep, 2025, doi:10.3390/pediatric17030053_

Round 1
Reviewer 1 Report
Comments and Suggestions for Authors
The pathology addressed, although rare, is impressive and could evolve unpredictably.
From my point of view, I think that in chapter 2 - methods, in paragraph 2 On physical examination.... the weight and the percentile on which it is located are correctly mentioned but the height is NOT specified.
In chapter 3 Results - I think a table with pre and post-operative values or pre/post hydration would be illustrative...
on page 3 of 9 before Figure 1 is 3.2.Figures, tables and Schemes ... which has no connection with either the text or the figure, it is explained in a photo
I also think that adding 5-6 bibliographic titles can add value to the work
Author Response
The pathology addressed, although rare, is impressive and could evolve unpredictably.
From my point of view, I think that in chapter 2 - methods, in paragraph 2 On physical examination.... the weight and the percentile on which it is located are correctly mentioned but the height is NOT specified.
1. Thank you for your comment. We added the patient’s height and BMI.
In chapter 3 Results - I think a table with pre and post-operative values or pre/post hydration would be illustrative…
1. Thank you for pointing this out. We agree with your comment. However no specific follow-up was done in those values.
on page 3 of 9 before Figure 1 is 3.2.Figures, tables and Schemes ... which has no connection with either the text or the figure, it is explained in a photo
1. Thank you for your comment, It has been removed.
I also think that adding 5-6 bibliographic titles can add value to the work
1. Thank you for pointing this out. We totally accepted your comments and did more literature research. New 5 references have been added.
Reviewer 2 Report
Comments and Suggestions for Authors
This article is interesting, and worth reading. Please consider and revise the following points.
Minor points
1)Keywords: Please avoid repetition, such as small bowel obstruction, and intestinal obstruction, acute abdomen and pediatric acute abdomen,---
2)Methods and Results: This is a case report. Please see “Instructions for Authors”.
3)Ultrasound or endoscopy was not performed?
4)Figures: Please add microscopic findings.
Author Response
Minor points
1)Keywords: Please avoid repetition, such as small bowel obstruction, and intestinal obstruction, acute abdomen and pediatric acute abdomen,---
Thank you for your comment, we removed unnecessarily repetitive keywords.
2)Methods and Results: This is a case report. Please see “Instructions for Authors”.
Thank you for pointing this out. The methods and results section is now converted into ‘Case Presentation’
3)Ultrasound or endoscopy was not performed?
Thank you for pointing this out. Ultrasound and endoscopy were not performed and that is one of the reasons we are trying to introduce this case, mainly to physicians in pediatric ED. When some of us are facing right lower quadrant pain, they almost suspect appendicitis without enough consideration because symptoms of appendicitis and related peritonitis are not typical in children compared to adults. And we tend to fear too much about misdiagnosis or losing the right time of the therapeutic window that may lead to other complications such as rupture of the appendix. But still, as we commented in the introduction section, it is important to understand the other conditions and not to overlook them. The section from the introduction is ‘Despite evaluating all the features mentioned, the possibility of overlooking a serious etiology is a major concern for physicians in pediatric ED. Acute appendicitis is the most common surgical cause of abdominal pain[2]; therefore, physicians tend to focus on certain physical indications, such as right-sided or rebounding tenderness, to rule out acute abdomen. However, it is important to understand that other rare conditions can be life threatening.’. We tried to add more about this in the discussion section.
4)Figures: Please add microscopic findings.
Thank you for pointing this out. The microscopic findings are added to the text.
Reviewer 3 Report
Comments and Suggestions for Authors
The case report and review is well-written and informative. However, I take issue with use of the term “Acute abdomen” and “surgical abdomen”. These terms imply an emergency that demands immediate surgical exploration; this clearly was not the case here.
I think that the authors are using the terms “acute abdomen” and “acute abdominal pain” interchangeably, and they are not the same thing.
Therefore, the title should be changed to something like, “Acute Abdominal Pain with Obstructing Trichobezoar: a case of Rapunzel Syndrome in a Child”
In addition, the term “acute abdomen’ should be removed elsewhere in the manuscript.
The word “fece” is misspelled in at least 2 places in the manuscript. It should be “feces”
“Although there was no evidence of extra trichobezoars on CT and ultrasonography, exploration for other possible residual trichobezoars was performed, and no evidence was found.”
This sentence is unnecessarily wordy. Just say, “No other trichobezoars were identified.”
I would include an axial image from the CT, and not just the coronal; this would assure the reader that there was no pneumoperitoneum and would also demonstrate the air-fluid levels indicating obstruction.
Fig 3: Really great pictures!
The legend is unnecessarily repetitive.
Change it to, “Figure 3. A 20cm U-shaped trichobezoar was found in the distal ileum, (b)The trichobezoar was removed after enterotomy.”
Comments on the Quality of English LanguageGenerally well written and good readability. Occasional wordiness.
Author Response
The case report and review is well-written and informative. However, I take issue with use of the term “Acute abdomen” and “surgical abdomen”. These terms imply an emergency that demands immediate surgical exploration; this clearly was not the case here.
I think that the authors are using the terms “acute abdomen” and “acute abdominal pain” interchangeably, and they are not the same thing.
1. Thank you for your comment. We agree that there are misuse of terms. We tried to eliminate or substitute the words ‘acute abdomen’ as much as possible. Thank you again.
Therefore, the title should be changed to something like, “Acute Abdominal Pain with Obstructing Trichobezoar: a case of Rapunzel Syndrome in a Child”
1. Thank you for pointing this out. We absolutely agree with your comment and the other reviewer has also pointed out that the title should be more refined. Therefore, We modified our title considering you and other reviewer’s suggestions. As you pointed out, we understand that the pathology is rare but well-documented in the previous literature. However, we would like to emphasize its pathology in the younger population, especially to the primary physicians working in pediatric ED. It is one of the reasons why we would like to put ‘pediatric emergency’ in the title. The authors kindly ask you to understand our intention.
In addition, the term “acute abdomen’ should be removed elsewhere in the manuscript.
1. Thank you for your comment. We agree that there are misuse of terms. We tried to eliminate or substitute the words ‘acute abdomen’ as much as possible. Thank you again.
The word “fece” is misspelled in at least 2 places in the manuscript. It should be “feces”
1. Thank you for this comment. They are corrected now.
“Although there was no evidence of extra trichobezoars on CT and ultrasonography, exploration for other possible residual trichobezoars was performed, and no evidence was found.”
This sentence is unnecessarily wordy. Just say, “No other trichobezoars were identified.”
1. Thank you for pointing this out. We changed the sentence as suggested. Thank you for your clear suggestion.
I would include an axial image from the CT, and not just the coronal; this would assure the reader that there was no pneumoperitoneum and would also demonstrate the air-fluid levels indicating obstruction.
1. Thank you for your comment. We included an axial image next to the coronal one. Thank you again.
Fig 3: Really great pictures!
The legend is unnecessarily repetitive.
Change it to, “Figure 3. A 20cm U-shaped trichobezoar was found in the distal ileum, (b)The trichobezoar was removed after enterotomy.”
1. Thank you for your clear and precise suggestion on this one. We changed the legend as suggested. We really appreciate it.
Reviewer 4 Report
Comments and Suggestions for Authors
-
The title should be evaluated for clarity and conciseness, ensuring it accurately reflects the key findings of the study.
-
The abstract could benefit from a clearer summary of the methodology, key results, and implications to provide a more structured overview of the study.
-
The background information should be more comprehensive, including a clearer rationale for why this study is necessary.
-
The objectives should be explicitly stated to highlight the research question and the hypothesis.
-
The methodology section should provide more details regarding patient selection criteria, sample size justification, and statistical methods used to ensure reproducibility.
-
Ethical considerations and approval should be explicitly mentioned, including any consent procedures followed.
-
The results section should include more comparative analysis to highlight significant findings.
-
Tables and figures should be clearly labeled and referenced within the text to facilitate comprehension.
-
The discussion should provide a stronger comparison with existing literature, addressing any discrepancies or reinforcing current knowledge.
-
The limitations of the study should be acknowledged more explicitly, with potential implications for future research.
-
The conclusion should concisely summarize the main findings and suggest clinical or research implications.
-
Any recommendations for future research should be clearly outlined to guide further investigations.
-
The manuscript should be reviewed for grammatical accuracy and clarity to enhance readability.
-
Sentences should be structured more concisely to avoid redundancy and improve overall flow.
Author Response
-
The title should be evaluated for clarity and conciseness, ensuring it accurately reflects the key findings of the study.
- Thank you for pointing this out. We absolutely agree with your comment and the other reviewer has also pointed out that the title should be more refined. Therefore, We modified our title considering you and other reviewer’s suggestions. As you pointed out, we understand that the pathology is rare but well-documented in the previous literature. However, we would like to emphasize its pathology in the younger population, especially to the primary physicians working in pediatric ED. It is one of the reasons why we would like to put ‘pediatric emergency’ in the title. The authors kindly ask you to understand our intention.
-
The abstract could benefit from a clearer summary of the methodology, key results, and implications to provide a more structured overview of the study.
- abstract has been modified
-
The background information should be more comprehensive, including a clearer rationale for why this study is necessary. The objectives should be explicitly stated to highlight the research question and the hypothesis,. The methodology section should provide more details regarding patient selection criteria, sample size justification, and statistical methods used to ensure reproducibility.
- This is case presentation, and we report our case according the journal's instructions for authors.
-
Ethical considerations and approval should be explicitly mentioned, including any consent procedures followed.
- The IRB number has been given. And the statement of the consent is written on the work.
-
The results section should include more comparative analysis to highlight significant findings.
- Thank you for pointing this out. We understand that, though it is a rare pathology but well-documented.
Our focus here is to show our diagnostic flow in pediatric trichobezoar patients with RLQ pain so that other physicians in pediatric ED can benefit from it.
We totally accepted your comments and did more literature research. Thank you again.
- Thank you for pointing this out. We understand that, though it is a rare pathology but well-documented.
-
Tables and figures should be clearly labeled and referenced within the text to facilitate comprehension.
- labels and legends are modified
-
The discussion should provide a stronger comparison with existing literature, addressing any discrepancies or reinforcing current knowledge.
-
Thank you for pointing this out. We totally accepted your comments and did more literature research. New 5 references have been added.
-
-
The limitations of the study should be acknowledged more explicitly, with potential implications for future research.
- Limitations such as failure to obtain the specific medical history and skipping an ultrasound are now mentioned.
-
The conclusion should concisely summarize the main findings and suggest clinical or research implications. Any recommendations for future research should be clearly outlined to guide further investigations.
- Conclusion is at the last of our work. The requirement for large numbers is mentioned about the future study suggestion in the conclusion section.
Reviewer 5 Report
Comments and Suggestions for Authors
The authors report the case of a 6-year-old girl who presented to the emergency department with acute abdominal pain and was diagnosed with Rapunzel syndrome.
While the pathology is rare, it is well-documented in the literature, and the current report does not introduce new findings. Below are my specific comments:
- The title is appropriate but could be refined slightly for clarity: “Acute abdomen due to trichophagia: a pediatric case of Rapunzel syndrome”
- The introduction is underdeveloped and does not meet academic standards. It should provide more background on trichobezoar, including presentation, epidemiology, diagnostic modalities, and treatment options.
- The authors did not even mention the treatment options that should be presented in the introduction. There are several options for treatment. I would suggest adding the following statement and an appropriate reference: The primary goal of treating bezoars is their removal and prevent recurrence. Several treatment options can be found in the literature, including removal by conventional laparotomy, laparoscopy and endoscopy (doi: 10.1080/00015458.2012.11680816).
- Case description – in addition to weight, height and BMI should also be reported.
- The authors stated that they suspected appendicitis or peritonitis and immediately performed laboratory tests and a computed tomography (CT) scan. Is this really standard of care in a 6-year-old pediatric patient with a soft and flat abdomen and a CRP of 1.05 mg/dl! Why didn't the authors perform an abdominal ultrasound instead of a CT scan? Obviously, this was not an emergency, so the correct diagnostic procedure should include an abdominal US scan, and if still in doubt, an MR scan could be scheduled. Next, the authors stated there was "suspicion of peritonitis" – I think this should be deleted as there was no evidence of peritonitis from the description of the physical exam and labs.
- Next question: since minimally invasive procedures should be standard today, why laparotomy instead of laparoscopy? This should be discussed and explained in the discussion.
- Authors should list all differential diagnoses originally considered and explain how they were excluded.
- The surgical procedure is described briefly but adequately. However, a section on postoperative follow-up could be more detailed. Was the recovery uneventful? What specific psychiatric support was initiated?
- The discussion is superficial and largely repeats known facts about trichobezoars. There is no meaningful analysis or integration of the findings into the existing body of knowledge. The authors should conduct a better literature search, include more relevant studies and compare these with their case.
- The quality of the English language is substandard. The manuscript should be revised by a native English speaker or a professional language editor to improve grammar and readability. Some parts of the manuscript contain grammatical and syntactical errors that affect readability.
- Finally, I see no benefit to readers from this report. Everything that is presented is well known and has been described many times in the literature. The authors should conduct a systematic review of the literature. Only in that way this manuscript can be of scientific value.
The quality of the English language is substandard. The manuscript should be revised by a native English speaker or a professional language editor to improve grammar and readability. Some parts of the manuscript contain grammatical and syntactical errors that affect readability.
Author Response
- The title is appropriate but could be refined slightly for clarity: “Acute abdomen due to trichophagia: a pediatric case of Rapunzel syndrome”
- Thank you for pointing this out. We absolutely agree with your comment and the other reviewer has also pointed out that the title should be more refined. Therefore, We modified our title considering you and other reviewer’s suggestions. As you pointed out, we understand that the pathology is rare but well-documented in the previous literature. However, we would like to emphasize its pathology in the younger population, especially to the primary physicians working in pediatric ED. It is one of the reasons why we would like to put ‘pediatric emergency’ in the title. The authors kindly ask you to understand our intention.
- The introduction is underdeveloped and does not meet academic standards. It should provide more background on trichobezoar, including presentation, epidemiology, diagnostic modalities, and treatment options.
- Thank you for this comment. We really appreciate this one.
- At the beginning of writing an initial draft, I personally was bothered to decide whether we should put more general background information in introduction or discussion. Since you also left a comment about lack of reviewing previous literature in the discussion section, I wrote background on trichbezoar’s presentation and epidemiology in the introduction and focused on its more detailed epidemiology, diagnostic modalities and treatment options in the discussion to avoid unnecessary repetition.
- We tried to perform more literature reviews and the new references and citations are also updated.
- The authors did not even mention the treatment options that should be presented in the introduction. There are several options for treatment. I would suggest adding the following statement and an appropriate reference: The primary goal of treating bezoars is their removal and prevent recurrence. Several treatment options can be found in the literature, including removal by conventional laparotomy, laparoscopy and endoscopy (doi: 10.1080/00015458.2012.11680816).
- Thank you for your comment and precise suggestion of the literature. We agree that it would be better to mention the treatment options. However, as we answered in the previous comment, we tried to focus on the condition’s treatment options in the discussion section. The suggested statement was added on the discussion section and the referenced has been added. We really appreciate it.
- Case description – in addition to weight, height and BMI should also be reported.
- Thank you for your comment. We added the patient’s height and BMI.
- The authors stated that they suspected appendicitis or peritonitis and immediately performed laboratory tests and a computed tomography (CT) scan. Is this really standard of care in a 6-year-old pediatric patient with a soft and flat abdomen and a CRP of 1.05 mg/dl! Why didn't the authors perform an abdominal ultrasound instead of a CT scan? Obviously, this was not an emergency, so the correct diagnostic procedure should include an abdominal US scan, and if still in doubt, an MR scan could be scheduled. Next, the authors stated there was "suspicion of peritonitis" – I think this should be deleted as there was no evidence of peritonitis from the description of the physical exam and labs.
- Thank you for pointing this out. We agree the ‘suspicion of peritonitis’ would not seem rational to readers. The quote has been deleted. However, the local clinic which referred the patient to us mentioned suspicion of appendicitis and peritonitis because of worsening right lower quadrant pain. Although there were not any signs of peritonitis, the pain got worse even under the treatment of the local clinic. We added ‘servere and unresolving’ for this matter.
- Our facility’s normal CRP value is under 0.3mg/dL, which has been now informed in the text.
- You definitely have a point and it is absolutely understandable to doubt whether the immediate CT scan was necessary or not. Actually, that’s why we are emphasizing the various reasons for the pediatric abdominal pain to the health workers in ED. When some of us are facing right lower quadrant pain, they almost suspect appendicitis without enough consideration because symptoms of appendicitis and related peritonitis are not typical in children compared to adults. And we tend to fear too much about misdiagnosis or losing the right time of the therapeutic window that may lead to other complications such as rupture of the appendix. But still, as we commented in the introduction section, it is important to understand the other conditions and not to overlook them. The section from the introduction is ‘Despite evaluating all the features mentioned, the possibility of overlooking a serious etiology is a major concern for physicians in pediatric ED. Acute appendicitis is the most common surgical cause of abdominal pain[2]; therefore, physicians tend to focus on certain physical indications, such as right-sided or rebounding tenderness, to rule out acute abdomen. However, it is important to understand that other rare conditions can be life threatening.’. Therefore it is very rational to have questions about our diagnostic flow, and it is actually one of our intentions for reporting this case. We tried to add more about this in the discussion section.
- When it comes to if the patient was not in emergent condition and considering an MR scan, the authors kindly disagree. First, the patient was referred for unresolving abdominal pain and she was actually in severe pain and going into dehydrated condition due to the intestinal obstruction. Secondly, even if her condition was not that emergent, we believe that the emergency medicine specialist was still obliged to categorize the patient’s condition so that the upcoming treatment could be most beneficial to her. In this case, finding out whether the pathology causing the abdominal pain is surgical or not seemed crucial. Therefore we believe US or CT scan would be more appropriate than MR scan in ED. An MR scan also would be more time-consuming and cost-ineffective. We understand your point, but kindly ask you to understand this patient was in an ER setting which was to decide if further treatment should be planned in Pediatrics or Pediatric Surgery.
- Next question: since minimally invasive procedures should be standard today, why laparotomy instead of laparoscopy? This should be discussed and explained in the discussion.
- Thank you for your comment. Yes, less invasive methods are preferred these days. We agree with that. Our pediatric surgeon is well experienced and has treated many trichobezoar patients before. He was afraid that there could be undetected satellite trichobezoars since it already passed gastric outlet and was very long in length. He performed laparotomy checking if there are left smaller trichobezoars in the small bowel manually. This is now discussed in the discussion section.
- Authors should list all differential diagnoses originally considered and explain how they were excluded.
- Thank you for pointing this out. We agree with your comment and we added other suspected pathology to the text. After the CT scan, which indicated bezoar like structure right away, there was not much to differentiate.
- The surgical procedure is described briefly but adequately. However, a section on postoperative follow-up could be more detailed. Was the recovery uneventful? What specific psychiatric support was initiated?
- Thank you for this comment. We understand that readers might wonder about the results of the postoperative status. We added brief details of the patient’s first follow-up. Since we were focused on introducing this pathology to the pediatric emergency physician, we were totally unaware about informing about the recovery and follow-ups. Thank you again.
- The discussion is superficial and largely repeats known facts about trichobezoars. There is no meaningful analysis or integration of the findings into the existing body of knowledge. The authors should conduct a better literature search, include more relevant studies and compare these with their case.
- Thank you for pointing this out. We understand that, though it is a rare pathology but well-documented. Our focus here is to show our diagnostic flow in pediatric trichobezoar patients with RLQ pain so that other physicians in pediatric ED can benefit from it. We totally accepted your comments and did more literature research. Thank you again.
- The quality of the English language is substandard. The manuscript should be revised by a native English speaker or a professional language editor to improve grammar and readability. Some parts of the manuscript contain grammatical and syntactical errors that affect readability.
- Thank you for pointing this out. We actually did submit the original draft to one of the most well known language editing services and got it revised. However, we absolutely understand that it could be low in quality of English language. We were informed that this journal provides language editing service so we decided to give it a try. Thank you.
- Finally, I see no benefit to readers from this report. Everything that is presented is well known and has been described many times in the literature. The authors should conduct a systematic review of the literature. Only in that way this manuscript can be of scientific value.
- Thank you for pointing this out. We understand that, though it is a rare pathology but well-documented. Our focus here is to show our diagnostic flow in pediatric trichobezoar patients with RLQ pain so that other physicians in pediatric ED can benefit from it. We totally accepted your comments and did more literature research. Thank you again.
Round 2
Reviewer 4 Report
Comments and Suggestions for Authors
The authors have responded satisfactorily to the reviewers' recommendations and have modified the manuscript accordingly. It can move forward in the publication process in the current version in my view.
Reviewer 5 Report
Comments and Suggestions for Authors
The revisions made by the authors are superficial. A discussion is still pointless. But even if the cut-off value of CRP is 0.3, CRP at 1.0 mg/dL is still extremely low. As I stated in my first review, I see no benefit to readers in this report. Everything presented here is well known and has been described many times in the literature. The authors should undertake a systematic review of the literature. Only then can this manuscript be of scientific value. The authors stated that they did a better literature review, but I suggested a SYSTEMATIC REVIEW according to the PRISMA guidelines. The authors did not do this, so this manuscript is of very low scientific quality.
Comments on the Quality of English Language--